# New Psychoactive Substances Consumption in Opioid-Use Disorder Patients

**DOI:** 10.3390/biology11050645

**Published:** 2022-04-22

**Authors:** Maria Alías-Ferri, Manuela Pellegrini, Emilia Marchei, Roberta Pacifici, Maria Concetta Rotolo, Simona Pichini, Clara Pérez-Mañá, Esther Papaseit, Robert Muga, Francina Fonseca, Marta Torrens, Magí Farré

**Affiliations:** 1Addiction Research Group, IMIM—Hospital del Mar Medical Research Institute, 08003 Barcelona, Spain; malias@imim.es (M.A.-F.); mffonseca@psmar.cat (F.F.); 2Department of Psychiatry and Legal Medicine, Universitat Autònoma de Barcelona, 08913 Cerdanyola del Vallés, Spain; 3National Centre on Addiction and Doping, Istituto Superiore di Sanità, 00161 Rome, Italy; manuela.pellegrini@iss.it (M.P.); emilia.marchei@iss.it (E.M.); roberta.pacifici@iss.it (R.P.); mariaconcetta.rotolo@iss.it (M.C.R.); simona.pichini@iss.it (S.P.); 4Department of Clinical Pharmacology, Hospital Universitari Trias i Pujol and Institut de Recerca Germans Trias i Pujol (HUGTiP-IGTP), 08916 Badalona, Spain; cperezm.mn.ics@gencat.cat (C.P.-M.); epapaseit.germanstrias@gencat.cat (E.P.); mfarre.germanstrias@gencat.cat (M.F.); 5Department of Pharmacology, Therapeutics and Toxicology, Universitat Autònoma de Barcelona, 08913 Cerdanyola del Vallès, Spain; 6Department of Internal Medicine, Hospital Universitari Germans Trias i Pujol and Institut de Recerca Germans Trias i Pujol (HUGTiP-IGTP), 08916 Badalona, Spain; rmuga.germanstrias@gencat.cat; 7Department of Medicine, Universitat Autònoma de Barcelona (UAB), 08913 Cerdanyola del Vallès, Spain; 8Addiction Program, Institute of Neuropsychiatry and Addictions, Hospital del Mar, 08003 Barcelona, Spain; 9Department of Medicine and Life Sciences (MELIS), Universitat Pompeu Fabra, 08003 Barcelona, Spain

**Keywords:** new psychoactive substances, opioid-use disorder, urine sample analysis

## Abstract

**Simple Summary:**

We applied a toxicological screening on 187 urine samples collected from patients with opioid-use disorder treated with opioid agonists in Barcelona and Badalona addiction care services, Spain. We found that 27.3% of urine samples were positive for any type of new psychoactive substance and 8.6% of samples were positive for a new synthetic opioid (NSO). These results show a new trend of consumption in patients with opioid-use disorder that requires social and political actions to stem associated health threats.

**Abstract:**

(1) Background: Since the beginning of the 21st century, the large number and wide chemical variety of new psychoactive substances (NPS) that enter the market every year has become a public health problem. Given the rapidity with which the drug market is changing, many NPS are not clinically investigated and their effects and health risks are unknown. Drug testing is a very useful tool for this purpose, but, unfortunately, it is not very widespread in individuals with opioid-use disorder under detoxification treatment. The aim of this study is to investigate the use of illicit drugs and NPS in opioid-use disorder (OUD) patients on opioid agonist treatment. (2) Methods: A multicenter, descriptive, cross-sectional study was conducted at two addiction care services in Barcelona and Badalona, Spain. Urine samples were collected from OUD individuals attending these two centers, who anonymously donated a urine sample at the time of a periodical visit. Samples were analyzed by high-sensitivity gas chromatography-mass spectrometry (GC-MS) and ultra-high-performance liquid chromatography-high –resolution mass spectrometry (UHPLC-HRMS). (3) Results: Out of the 187 collected and analyzed urine samples, 27.3% were positive for any type of NPS and 8.6% were positive for new synthetic opioids, including fentanyl and its derivatives (NSO). Other frequently detected substances were benzodiazepines in 46.0% of samples, antipsychotics in 27.8% of samples, or cocaine and cannabis in 23.5% of samples. (4) Conclusion: A wide number of NPS, including NSO, have been detected in urine samples from an OUD population. A lack of NPS detection in standard drug screening among drug users can hide the identification of a potential public health problem.

## 1. Introduction

In the 20th century, the illicit market of drugs of abuse was limited to a few classes of psychotropic substances such as cannabis, opiates, cocaine, amphetamines, hallucinogens, and benzodiazepines [1]. In the 21st century, that market expanded exponentially with the availability of new psychoactive substances (NPS), a very heterogeneous group of substances with a wide range of mechanisms and chemical variety [2,3,4].

Worldwide, 1124 NPS have been reported to the UNODC Early Warning Advisory from 2009 up to January 2022 [5], and in Europe, in these first twenty years of the 21st century, more than 1000 NPS [6] were made available on the street and internet dealing. Some of these entered and left the illicit market very quickly, while others persisted over the years and some showed a sharp demand increase, especially during the COVID-19 pandemic [7,8]. Although in Europe, the NPS most commonly detected in both drug seizures and intoxications are synthetic cannabinoids and synthetic cathinones, in recent years, a rapid and constant barrage of New Synthetic Opioids (NSO) has been observed [9].

Polysubstance use typically includes the simultaneous consumption of three or four psychotropic substances from opiates, cocaine, cannabinoids, and amphetamines classes [10]. More recently, polyconsumption can also involve the addition of NPS [11,12]. However, it is yet unclear how many users of “classical” illicit psychotropic substances are attracted by NPS. Along with this, some NPS have been used as street opiates adulterants, being fentanyl and their analogues as the most common ones [13]. This latter occurrence entails a risk for users, either because of a lack of knowledge of the consumed product and/or because of the high potency of the above-reported adulterants, which can result in a fatal overdose [14]. In general, NPS users are young individuals who also use other substances in a recreational setting, usually did not have a concomitant substance use disorder [15], and are the most frequently intoxicated by the use of these substances [16]. One exception can be the use of NSO, such as fentanyl and its derivatives, that are most commonly abused by subjects with an opioid-use disorder (OUD) [17].

The screening of NPS in clinical practice is not widespread, as well as in recreational consumers [18]. Drug checking is a very useful strategy for harm reduction, as well as a way to identify substances available on the market [3]. This service usually offers consumers the possibility to analyze illicit drugs before consumption, whereas post-consumption drug checking, which allows one to know the substances consumed, is less common [19].

Several studies have investigated the use and/or detection of classical psychoactive drugs and NPS in mainly recreational users (e.g., raves, musical festivals, etc.) [20], through wastewater analysis [21], in emergency rooms when intoxication, overdose, or death was attributed to the use of these substances [22], or in patients in detoxification treatment [23]. Fewer studies have focused on individuals with OUD on opioid maintenance treatment. This population presents with a high prevalence of polysubstance use, including NPS among the abused compounds [23]. Apart from these studies, the consumption of NPS in populations with OUD is scarcely studied in some European regions and specifically in Mediterranean areas.

In this regard, we investigated the consumption of common drugs of abuse and NPS in individuals with OUD attending outpatient addiction care services in the greater Barcelona area (Barcelona city and Badalona, Spain) by urinalysis.

## 2. Materials and Methods

### 2.1. Study Design and Participants

A multicenter, descriptive, cross-sectional study was conducted at the addiction care services of the Hospital del Mar, Barcelona, Spain and the Hospital Universitari Germans Trias i Pujol, Badalona, Spain, from February 2019 to March 2020 and from July to October 2020. Due to the COVID-19 pandemic and the impact it has caused on the functioning of addiction care services, sample collection was cancelled from 13 March until 6 July 2020.

The subjects enrolled in the study donated an anonymous urine sample during their regular urine test at the addiction care service and, since the participation was voluntary and anonymous, their personal data or any other medical information were not recorded. Subjects had to meet the following inclusion criteria: being over 18 years of age, having an opioid-use disorder according to DSM-5 criteria [24], and being on opioid agonist treatment. No exclusion criteria were applied.

The study was approved by the Ethics Committee in Clinical Research Parc de Salut MAR (CEIC-PSMAR, number 2018/8138/I) and Hospital Universitari Germans Trias i Pujol (CEIC-HUGTiP number PI-18-126).

### 2.2. Urinalysis

Urine samples from recruited individuals were collected without any preservative and stored at −20 °C until analysis. Urinalysis was performed by two different set-ups and validated methodologies. An ultra-high-performance liquid chromatography-high-resolution mass spectrometry (UHPLC-HRMS) assay was used for extensive screening of more than 1000 pharmacologically active substances, including prescription psychoactive drugs, classic drugs of abuse (e.g., opiates, cocainics, amphetamine-type substances, cannabinoids, hallucinogens, etc.), NPS (parent drugs and metabolites), prescription opioids (e.g., oxycodone, hydromorphone, hydrocodone, etc.), NSO such as fentanyl and analogs, and benzoimidaloles (e.g., etonitazene, isotonitazene and metonitazene) [25]. A last generation gas chromatography-mass spectrometry method was used for the confirmation of identified compounds [25].

### 2.3. Data Analysis

The rates for each detected substance and metabolites were described as frequencies and percentages using the SPSS version 22.0 (SPSS Inc., Chicago, IL, USA) software.

## 3. Results

One hundred eighty-seven participants were recruited for the study and donated a urine sample. Although the samples were collected anonymously, they are part of an opioid agonist treatment program (main characteristics: 68% men, mean age 52 years old, range: 28–77). The NPS detections are shown in Table 1 and the detection of other substances in Table 2.

Some type of NPS (opioid, stimulant, or cannabinoid) was detected in 51 (27.3%) of the 187 urine samples and a total of 45 different NPS were detected (Table 1). In addition, more than one substance was detected in 124 (66.3%) urine samples.

Fentanyl and derivatives were present in the urine of 16 (8.6%) participants and only one of these samples was positive for heroin too. Stimulant-type NPS were detected in 35 (18.7%) participants, being 4-Methyl-PV8 (*n* = 7, 3.7%) and m-CPP (*n* = 7, 3.7%) as the most detected substances, followed by 1-(4-chlorophenyl) piperazine (*n* = 4, 2.1%). In addition, seven of these subjects were also positive for cocaine (25%). Cannabinoid-type NPS were detected in six (3.2%) participants, with the most detected being was JWH-122 (*n* = 4, 2.1%).

In agreement with the administered treatment, an opioid agonist was detected in the majority of samples (*n* = 177, 94.6%): methadone in 174 (93.0%) participants, morphine in two (1.1%) participants, and buprenorphine in one (0.5%) participant (Table 2). Benzodiazepines (*n* = 86, 46.0%) were the most frequently detected psychiatric treatment drugs, followed by antipsychotics (*n* = 52, 27.8%) and antidepressants (*n* = 50, 26.7%). In 43 (23.0%) participants, an anticonvulsant was found and the main one was gabapentin (*n* = 20, 10.7%), followed by pregabalin (*n* = 7, 3.7%). Non-opioid analgesics were detected in 27 (14.4%) participants. Stimulants were present in 49 (26.2%) samples and the majority were positive for cocaine (*n* = 44, 23.5%). Less commonly detected was amphetamine or methamphetamine, being positive in two (1.1%) samples each. Opioids were detected in a total of 30 (16%) participants. Among these, most detected opioids were dextromethorphan and heroine, being positive in 11 (5.9%) samples each. Finally, other detected drugs were alcohol in 17 (9.1%) samples, metabolites of THC (11-COOH-THC) in 44 (23.5%), and LSD/LAMPA in three (1.6%) samples.

## 4. Discussion

Different types of NPS, other substances of abuse, and psychiatric and other treatment drugs have been detected in our study. We detected the presence of any type of NPS in 27.3% of urine samples from patients with an OUD diagnosis attending a treatment centre. Differentiating by type of NPS, we detected NSO and/or fentanyl in 8.6% of the samples, NPS stimulant type in 18.7%, NPS cannabinoid type in 3.2%, and other NPS in 1.6% of samples. Additionally, opioids other than NSO were found in 16% of our samples.

Specifically, the presence of fentanyl in our samples agree with what was previously highlighted in one of our previous studies (6.1% versus 8.6%) [17]. Another study conducted with opioid maintenance treatment shows a 13.0% prevalence of NPS use, although, unlike our results, no fentanyl and analogs or NSO was found [26]. Similar to these data, NSOs were also not found in the substance-use-disorder population despite subjects having reported their use [23]. These findings may explain the differences in the prevalence of NPS and NSO use between studies.

According to worldwide NPS identifications [2], most detected in our samples was stimulant-type NPS. However, our results showed a higher use of NSO in this population than expected based on drug seizure data.

The prevalence for NPS use in Europe is 1.1% among young adults (15–34 years old) [6], although its use is normally associated with another substance such as alcohol, cocaine, or heroin [27]. In our sample, 22.9% of those individuals consuming an NPS stimulant-type also used cocaine, the use of which is widespread among people receiving opiate substitution treatment [28].

Other combinations of substances detected were with heroin, which was present in 6.3% of the samples positive for fentanyl and in 20.8% of the samples positive for other opioids. This is in agreement with the reported heroin adulteration with NSO and other opioids with increased addictive potency and risk of unintended intoxication for heroin users [29]. Moreover, there is a high proportion of polysubstance use consumption among people with an OUD [26] and is often addressed to the classic prescription opioids and NSO [19,30].

We found an elevated consumption of psychiatric treatment drugs such as benzodiazepines, antipsychotics, antidepressants, and anticonvulsants in our sample of OUD patients, as described commonly in other similar studies. The possible biological role of this high prevalence of psychiatric drug use is probably related to a dual diagnosis including psychosis, affective disorders, anxiety, and personality disorders [31,32,33]. Estimating the prevalence of use of NPS and NSO is complicated for several reasons: the non-detection of these substances in standard toxicological tests [4], the unawareness of their use by consumers [34], and the continuous change in the drug market [35]. In addition, as in our case, many of the NPS detected have not previously been described in the scientific literature, so their mechanism of action, effects, and health risks are unknown.

One relevant difference between young recreational NPS users and our population is that people with an OUD are not always aware of drug-checking services or are not sufficiently motivated to bring their substances for testing. These circumstances point to the post-consume drug checking as a suitable tool in the OUD population. This technique allows us to get an overview of all the substances being consumed, voluntary or involuntary, and detect substances missed by ordinary controls. The importance of drug-checking as a harm reduction tool in the clinical setting should be emphasized, not only in recreational contexts [19].

In the last 10 years, the use of NPS, as well as NSO, has been consolidated as a global health problem. Hence, new public health and social measures are needed, including the development of detection methods for these substances, early detection strategies, as well as specific prevention and treatment strategies [36].

## 5. Limitations

This study has some limitations: (I) the most relevant is the sample under investigation, which selected between participants who want to collaborate in the research and was not a random sample; (II) the anonymous collection cannot permit one to know the origin of the opioids (prescription and/or illegal market); (III) the design of our study is cross-sectional and subjects were not followed for a period; (IV) differences between gender or ages cannot be evaluated due to study design. The possibility of substance detection will depend on the time of consumption, dose, and elimination half-life in urine in relation to the sample collection.

## 6. Conclusions

We detected a wide variety of NPS of different types in a sample of patients with an OUD. The detection of NSO and other opioids in our sample suggests a non-therapeutic use of these substances. Difficulties in analyzing NSO and NPS in urine samples makes it difficult to extend the knowledge of the use of these substances in opioid treatment centers. It is necessary to follow up the NPS phenomenon in different populations of drug users through its detection in urine and other biological matrices.

## Figures and Tables

**Table 1 biology-11-00645-t001:** Detected NPS and metabolites in biological samples (N = 187).

NPS	Detected in Urine, *n* = 187
NSO AND FENTANYL	16 (8.6)
2F-ortho-fluorofentanyl	1 (0.5)
2-Fluorofentanyl	1 (0.5)
Acetyl-methyl fentanyl	1 (0.5)
Beta-hydroxyfentanil	2 (1.1)
Fentanyl	7 (3.7)
Fluorofentanyl	2 (1.1)
Fluoro valeril fentanyl	1 (0.5)
Meta fluoro valeril fentanyl	3 (1.6)
Norfentanyl	7 (3.7)
Thiofentanyl	2 (1.1)
NPS STIMULANT TYPE	35 (18.7)
1-3-chlorophenyl piperazine	3 (1.6)
1-(4-chlorophenyl) piperazine	4 (2.1)
2,fluorophenyl piperazine	1 (0.5)
2,4 Dimethoxyamphetamine	1 (0.5)
25N BoMe	1 (0.5)
3.4 methylendioxypyrovalerone	2 (1.1)
3.4 methylendioxy PV8	1 (0.5)
4-cloro N butyl cathione	1 (0.5)
4-Fluoro-PV8	1 (0.5)
4-Fluoroamphetamine	1 (0.5)
4-Methoxy-PV8	1 (0.5)
4-Methyl-PV8	7 (3.7)
5-AEDB	1 (0.5)
B-Methylphenethylamine (BMPEA)	1 (0.5)
BK-MPA	1 (0.5)
Buphedrone	2 (1.1)
Dimethylcathione	1 (0.5)
Ephinidine	1 (0.5)
Fenethylline	2 (1.1)
Lefetamine	1 (0.5)
m-CPP (1-(3-chlorophenyl)piperazine)	7 (3.7)
Methoxyamphetamine	2 (1.1)
Methoxyphenedine	1 (0.5)
MD-Benzyl MDMA	1 (0.5)
Methylendioxypyrovalerone (MDPV)	1 (0.5)
Ortho-chlorophenylpiperazine	3 (1.6)
NPS CANNABINOID TYPE	6 (3.2)
JWH-018	1 (0.5)
JWH-032	1 (0.5)
JWH-122	4 (2.1)
JWH-122 N-4-hydroxypentyl / JWH-122 N-5-hydroxypentyl	3 (1.6)
JWH-200	1 (0.5)
JWH-210	2 (1.1)
JWH-210 N-4-hydroxypentyl / JWH-210 N-5-hydroxypentyl	2 (1.1)
UR-144	1 (0.5)
UR-144 N-5-hydroxypentyl	1 (0.5)

**Table 2 biology-11-00645-t002:** Other detected substances and metabolites in biological samples (N = 187).

Other Substances	Detected in Urine, *n* = 187
OPIOID SUBSTITUTION DRUGS	
Methadone	174 (93)
Morphine	2 (1.1)
Buprenorphine	1 (0.5)
PSYCHIATRIC TREATMENT DRUGS	
Antidepressants	50 (26.7)
Antipsychotics	52 (27.8)
Anticonvulsant	43 (23.0)
Benzodiazepines	86 (46.0)
OTHER THERAPEUTIC DRUGS	
Non-steroidal anti-inflammatory	6 (3.2)
Non-opioid analgesic	27 (14.4)
Anesthetic	2 (1.1)
Non-opioid alkaloid	2 (1.1)
Anesthetic (Lidocaine)	2 (1.1)
Other drugs *	43 (23)
STIMULANTS	49 (26.2)
Cocaine	44 (23.5)
Amphetamine	2 (1.1)
Ephedrine	2 (1.1)
Ethylamphetamine	3 (1.6)
Feprosidnine	1 (0.5)
Methamphetamine	2 (1.1)
Norephedrine	1 (0.5)
OPIOIDS	30 (16)
Heroin	11 (5.9)
Alfa-propoxyphene	1 (0.5)
Codeine	7 (3.7)
Desmethyltramadol	2 (1.1)
Dextromethorphan	11 (5.9)
Hydromorphone	3 (1.6)
Levophanol/Dextrorphan	6 (3.2)
Norcodeine	1 (0.5)
Norpropoxyphene	2 (1.1)
Oxymorphone ether TMS	1 (0.5)
Tramadol	1 (0.5)
OTHER DRUGS	
Alcohol	17 (9.1)
11-Nor-9-carboxy-Δ9-tetrahydrocannabinol (11-COOH-THC)	44 (23.5)
LSD/LAMPA	3 (1.6)

* Levamisol, azapetine, etidronate, bisopropol, domperidone, furosemide, enalapril, etidronate.

## Data Availability

Data is contained within the article.

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
