# Peer review of "New Psychoactive Substances Consumption in Opioid-Use Disorder Patients"

_biology, 2022, doi:10.3390/biology11050645_

Round 1
Reviewer 1 Report
Alias-Ferri et al. investigated the new psychoactive substances consumption in opioid use disorder patients by performing a toxicological screening on 187 urine samples from Spain. They found that more than 27% urine samples were positive to any type of new psychoactive substances (NPS) and about 8 % samples were positive to the new synthetic opioids (NSO).
The authors used an UHPLC-HRMS assay to screen more than 1,000 drugs, then further performed a GC-MS to confirm the identified compounds. The authors performed a lot of works, however, throughout the manuscript, could not find the impact of the work. I would suggest the authors emphasize these in the results section, additionally if the authors have the UHPLC-HRMS or GC-MS data graph, they could add it into the results.
Another comment is that the authors could add the gender, age, and general health condition into the table, which can provide more detailed information to the readers.
Reviewer 2 Report
Comments attached

Reviewer 3 Report
The authors conducted a study of interest from a scientific and practical point of view. However, the study is predominantly clinical, but not biological. The Introduction section should be supplemented with up-to-date information on the biological role of the studied psychoactive substances.
Technical remarks:
- I recommend deleting Simple Summary, leaving only Abstract.
- It is necessary to fill in the Citation section.
- Add the research goal to the Abstract.
- Materials and methods: formalize the criteria for inclusion and exclusion of participants; describe the methods of statistical analysis of the database.
- Results: I recommend in the text to leave only the specific weight (%) of cases of detection of psychoactive substances in urine, removing the absolute numbers.
- Move the limitations of the study to a separate section Limitations after the Discussion section.
- Discussion: Please add a table to compare the results of your study conducted in Spain with similar studies in other European countries or in the world.
- References: must be designed in accordance with the requirements of the journal (see: guidelines for authors).
Author Response
Please see the attatchment.

Round 2
Reviewer 1 Report
The authors answered all my comments, I do not have further questions.
Reviewer 2 Report
The responses to the comments were generalized and there was no specific answer to comments 4,5 and 6. Overall, there are no significant improvements shown in data and results in analysis and interpretation. Since there is no novel findings were established in this revised paper and therefore this manuscript might not be suitable as an original research article, though the editor can consider it as a communication/review article.
Reviewer 3 Report
Dear Authors,
Thank you for modifying your manuscript.
The article may be recommended for publication at the discretion of the editor.